# Association between physical multimorbidity and common mental health disorders in rural and urban Malawian settings: Preliminary findings from Healthy Lives Malawi long-term conditions survey

Owen Nkoka[1,2]*, Shekinah Munthali-Mkandawire[1], Kondwani Mwandira[1], Providence Nindi[1], Albert Dube[1], Innocent Nyanjagha[1], Angella Mainjeni[1], Jullita Malava[1], Abena S. Amoah[1,3,4], Estelle McLean[1,3], Robert C. Stewart[1,5], Amelia C. Crampin[1,2,3], Alison J. Price[1,3]

1 Malawi Epidemiology and Intervention Research Unit, Lilongwe, Malawi, 2 School of Health and Wellbeing, University of Glasgow, Glasgow, United Kingdom, 3 Department of Population Health, London School of Hygiene and Tropical Medicine, London, United Kingdom, 4 Department of Parasitology, Leiden University Medical Centre, Leiden, The Netherlands, 5 Division of Psychiatry, Centre for Clinical Brain Sciences, College of Medicine and Veterinary Medicine, University of Edinburgh, Edinburgh, United Kingdom

* onkoka@gmail.com

## Abstract

In low-income Africa, the epidemiology of physical multimorbidity and associated mental health conditions is not well described. We investigated the multimorbidity burden, disease combinations, and relationship between physical multimorbidity and common mental health disorders in rural and urban Malawi using early data from 9,849 adults recruited to an on-going large cross-sectional study on long-term conditions, initiated in 2021. Multimorbidity was defined as having two or more measured (diabetes, hypertension) or self-reported (diabetes, hypertension, disability, chronic pain, HIV, asthma, stroke, heart disease, and epilepsy) conditions. Depression and anxiety symptoms were measured using the 9-item Patient Health Questionnaire (PHQ-9) and the 7-item General Anxiety Disorder scale (GAD-7) and defined by the total score (range 0–27 and 0–21, respectively). We determined age-standardized multimorbidity prevalence and condition combinations. Additionally, we used multiple linear regression models to examine the association between physical multimorbidity and depression and anxiety symptom scores. Of participants, 81% were rural dwelling, 56% were female, and the median age was 30 years (Inter Quartile Range 21–43). The age-standardized urban and rural prevalence of multimorbidity was 14.1% (95% CI, 12.5–15.8%) and 12.2% (95% CI, 11.6–12.9%), respectively. In adults with two conditions, hypertension, and disability co-occurred most frequently (18%), and in those with three conditions, hypertension, disability, and chronic pain were the most common combination (23%). Compared to adults without physical conditions, having one ($B$-Coefficient ($B$) 0.79; 95% C1 0.63–0.94%), two- ($B$ 1.36; 95% CI 1.14–1.58%), and three- or more- physical conditions ($B$ 2.23; 95% CI 1.86–2.59%) were associated with increasing depression score, $p$-trend <0.001. A comparable 'dose-response' relationship was observed between physical

**Data Availability Statement:** The anonymized data relevant to this study can be obtained by submitting a reasonable request to the Malawi Epidemiology and Intervention Research Unit. For data inquiries, please contact info@meiru.mw, quoting the title of this paper.

**Funding:** This work is supported by the Wellcome Trust, awarded to ACC (217073/Z/19/Z). The funder has no involvement in the study design, data collection, analysis, decision to publish, or the preparation of the manuscript. All aspects of the research have been conducted independently and are solely the responsibility of the authors.

**Competing interests:** The authors have declared that no competing interests exist.

multimorbidity and anxiety symptom scores. While the direction of observed associations cannot be determined with these cross-sectional data, our findings highlight the burden of multimorbidity and the need to integrate mental and physical health service delivery in Malawi.

## Introduction

In low- and middle-income countries (LMIC), population life expectancy has improved in recent decades (12 year increase in life expectancy at birth observed between 1990 and 2015 [1, 2]), in part due to greater healthcare access and higher living standards [3, 4]. In LMIC of Africa with generalised HIV-infection epidemics, the widespread availability of free, at point of care, highly effective antiretroviral medication for HIV infection has played a crucial role [5] in prolonging the lives of individuals living with HIV, who now survive to experience other health conditions (that may also be exacerbated by long-term treated HIV), such as dyslipidaemia, hypertension and diabetes [6]. As such, the burden of people living with multiple long-term health conditions is increasing, with implications for individual wellbeing and wider societal and economic impacts [7].

Multimorbidity, defined as the presence of two or more long-term conditions [8], is a concern for healthcare systems worldwide [9, 10]. In recent studies, the reported prevalence of multimorbidity in LMIC adults vary widely, with estimates ranging from 3.2% to 90.5% [11]. While the bulk of the multimorbidity burden is observed in those aged ≥60 years, emerging research shows multimorbidity even in younger adults (aged 18–49 years) [10, 12–14], with a prevalence of 5.6% observed in Botswana in individuals aged 15–44 years [12]. The observed heterogeneity in multimorbidity prevalence estimates in different populations [14] and emerging burden in young adults highlights the need for, country-specific studies to determine high-risk and vulnerable groups, socioeconomic and lifestyle factors that influence burden [10] and effective early detection and intervention strategies that mitigate the long-term health impacts.

Mental health disorders in LMIC represent a substantive health burden, accounting for 80% of people with mental health conditions worldwide [15]. However, 90% of individuals with mental health disorders in LMIC do not have access to mental health services [16–18]. This treatment gap is further exacerbated by inequalities in accessing mental health services between rural and urban residents [19]. Previous studies have reported links between mental health disorders and multimorbidity [20, 21]. In rural and urban India, compared to those with one or no health conditions, multimorbid individuals exhibited a higher risk of depressive symptoms measured using the Centre for Epidemiologic Studies Depression Scale (CES-D) [22].

In Malawi, a low-income, high HIV prevalence (8.9%) [23] African country, the primary healthcare system is under-resourced and based largely on a single-disease delivery paradigm [24] despite a rising burden of non-communicable diseases (NCDs)–including hypertension, diabetes and chronic respiratory conditions [25, 26]. To date, several facility-based studies in Malawi have reported a high burden of mental health disorders [27, 28]. Nonetheless, most mental health conditions remain under diagnosed in the community due to limited availability of mental health services in primary care, and a lack of mental health funding and trained mental health professionals [29].

The relationship between multimorbidity and mental health disorders may be biological as well as social. For example, elevated inflammatory markers and stress, have been found to be

associated with an increased risk of developing depression [30, 31] as well as with physical long-term conditions [32]. Epidemiological studies have also established links between depression, anxiety, and cardiovascular diseases [33–35].

In LMIC, there is some evidence of the impact of individual physical health conditions on mental health [36–38] yet, few studies have investigated the relationship between multiple physical health conditions and depression and anxiety. A population-based study in South Africa observed a positive association between multimorbidity, defined as hypertension and one other chronic physical condition, and anxiety [39] that was not observed for the association of hypertension alone and anxiety. A better understanding of this complex relationship is needed to provide holistic patient care and to develop effective interventions for the prevention and management of these conditions.

In this study we investigated the burden and common combinations of multiple health conditions, and the association between physical multimorbidity and common mental health disorders using early data from an ongoing population based cross-sectional study of long-term conditions in adolescents and adults being conducted in rural and urban Malawi between 2021 and 2024.

## Methods

### Study design

The long-term conditions study in rural and urban Malawi is an ongoing cross-sectional survey gathering information on burden of conditions and their risk factors, along with anthropometric and biological measurements. We aim to recruit 50,000 adolescents (15+ years) and adults who are resident in rural and urban health and demographic surveillance systems (HDSS), which operate as open cohorts. The survey is one component of a wider "Healthy Lives Malawi" research platform, investigating long-term health conditions using population-level data from the rural and urban HDSS.

### Study setting

The rural Karonga HDSS was started in 2002 and currently captures information on the births, deaths and movements of approximately 53,000 people covering an area of 140km$^2$ [40]. It is situated in the northern part of Malawi, in Karonga district, with a predominantly subsistence economy [40]. The Lilongwe HDSS (urban) covers an area of 6.68 km$^2$ and is nested within Area-25, a high-density area located in the west of Lilongwe city that borders 'Kanengo', an industrial location including tobacco auction floors. Baseline recruitment for the Lilongwe HDSS started in August 2022. At the time of analysis, over 7,000 of the 65,000 people targeted by the Lilongwe HDSS have been recruited and are being followed. The population includes people working in public and private sectors, including many seasonal tobacco workers due to the location's proximity to the Kanengo industrial area [41].

### Study population

Individuals aged 15 years and above, who were usual members/residents of households in the rural or urban HDS were eligible for recruitment. Usual members were defined as individuals who live in a household within the surveillance areas for a significant portion of the year and consider it their primary place of residence. We recruited adolescents to understand better the early risk factors for long-term conditions and their developmental origins, given that the previous survey [42] already identified measurable prevalence of obesity, hypertension and diabetes at age 18 years.

## Data collection

Data collection was interviewer-led using electronic data capture with Open Data Kit software [43]. Questions were read to each participant in the language of their choice. Most interviews in Karonga were conducted in Chitumbuka, those in Lilongwe were mostly in Chichewa. All field workers were trained using standard operating procedures, including obtaining informed, written (or a thumbprint for individuals unable to write) consent from participants, biomeasures and venepuncture. A pre-test of the survey tool and study procedures was carried out from July 26 to August 06, 2021 (in rural site Karonga) and from July 18 to July 29, 2022 (in urban site Lilongwe). Data collection in Karonga commenced in October 2021, and Lilongwe in September 2022.

## Definitions

**Main outcomes.** *Depression.* We used Chichewa and Chitumbuka versions of the Patient Health Questionnaire-9 (PHQ-9), developed through a process of translation, adaptation, and piloting, and validated against clinical interviews, to evaluate symptoms of depression [44]. The PHQ-9 is a survey measure that includes items that enquire about the presence of symptoms of Diagnostic and Statistical Manual of Mental Disorders (DSM) depressive disorder in the previous two weeks. In the versions of PHQ-9 used in this study, each item is scored from '0' (not at all), "1" (sometimes), "2" (usually), and "3" (almost always). The PHQ-9 scores range from 0 to 27, with higher scores corresponding to more symptoms and increasing severity of depression [45].

*Anxiety.* We used Chichewa and Chitumbuka versions of the General Anxiety Disorder (GAD)-7 scale [46], developed through a process of translation, adaptation, and piloting, and validated against clinical interviews, to evaluate symptoms of anxiety. The GAD-7 is a seven-item survey measure that assesses for symptoms of DSM generalized anxiety disorder in previous two weeks. In the versions of the GAD-7 used in this study, each item is scored from '0' (not at all), "1" (sometimes), "2" (usually), and "3" (almost always). The GAD-7 scale has a score range of 0 to 21, with higher values indicating more symptoms and more severe anxiety [46].

**Main independent variable.** Physical health multimorbidity was defined as the presence of two or more long-term conditions [8]. These included a self-report of previously clinician-diagnosed conditions such as heart disease, asthma, epilepsy, stroke, and HIV. A question 'have you ever been diagnosed with [*name of condition*] by a doctor or other health professionals' was asked for each of the conditions to obtain self-reported medical history of the participants. The Washington Group short set of questions on disability's six functional domains—difficulty seeing even with glasses, difficulty hearing even with a hearing aid, difficulty walking or climbing stairs, difficulty remembering or concentrating, difficulty washing oneself thoroughly or dressing—were used to capture disability [47]. Disability was defined as the inability to perform any of the six disability domains indicated by responses "a lot of difficulty" or "can't do at all." Participants who reported having any pain for more than three months were categorised as having chronic pain [48] with information on location also collected. Diabetes was defined as fasting blood glucose of ≥7.0 mmol/L [49] or self-report of a prior diagnosis of diabetes by a health professional. A systolic blood pressure of ≥140 mm Hg, a diastolic blood pressure of ≥90 mm Hg [50], or self-reporting of current antihypertensive drug use were all used to define hypertension. Individuals with incomplete data for whom it was not possible to determine if they had at least 2 conditions, only 1 or no conditions, were excluded from the multimorbidity definition.

**Covariates.** Covariates included: site (Lilongwe (urban), Karonga (rural)); age in years (15–17, 18–29, 30–39, 40–49, 50–59, and ≥60); sex (male vs female); education attainment

(not completed any level, primary, junior secondary, senior secondary, and postsecondary education); employment (not employed, full-time student, regular employment, and self/irregular employment); and marital status (never married, married, divorced/separated, and widowed). We also considered several lifestyle and other health-related factors such as smoking status (never smoked, current [smoked in the last 6 months], and former [stopped more than 6 months ago]); alcohol consumption in the past one year (yes/no); being physically active as defined by report of activities that strengthen muscle at least 3 days per week (for those aged 15–17 years) and at least 2 days per week (among those aged ≥18 years) [51]; and body mass index (BMI) measured as weight in kilograms by height in meters squared (kg/m$^2$) based on WHO conventional classification and categorized into four groups (underweight < 18.5, normal 18.5–24.9, overweight 25.0–29.9 kg/m$^2$ and obesity ≥30.0 kg/m$^2$) [52].

## Data analysis

We performed univariate and bivariate analyses (using one-way analysis of variance (ANOVA) or t-tests, as appropriate) to describe the main variables and the relationship between selected factors and depressive and anxiety symptoms as well as multimorbidity status. To account for multiple testing, we also performed a Bonferroni test on associations for which the ANOVA p value was < 0.05. Age-standardized prevalence of the physical and mental conditions considered in the current analysis and the mental health measures were calculated using WHO standard population [53] to allow meaningful comparison with external populations.

To investigate the association between physical multimorbidity and depressive and anxiety symptoms, we applied multiple linear regression models without accounting for clustering of individuals from the same households (as interclass correlation coefficients (ICC) of responses for participants from the same household in relation to depressive and anxiety symptoms were low (ICC 0.14 and 0.09, respectively)). We tested for trend to determine whether there was a 'dose-response' relationship between increasing number of physical conditions and depression and anxiety scores [54, 55].

We investigated the extent to which the association between physical health and multimorbidity varied by key sociodemographic factors (sex, site, and age), by adding an interaction term to the regression model one at a time. We further conducted stratified analyses for variables with significant interactions ($p<0.05$) as this signalled potential moderation effects.

We used $\chi^2$, t-tests and ANOVA to compare the sociodemographic characteristics (sex, age, educational level, employment) and the outcome measures in the individuals that were omitted from the analysis due to incomplete data on multimorbidity status to those with complete data to check whether there were any significant differences.

All analyses were performed using Stata version 17.0 (Stata Corp LP, College Station, TX, USA), and a 2-tailed $p$-value < 0.05 was considered significant.

## Ethical considerations

The Healthy Lives Malawi long-term conditions survey protocol was reviewed and approved by the Malawi National Health Sciences Research Committee (approval #2642) and the University of Glasgow's College of Medical, Veterinary and Life Sciences Ethics Committee (project #200200057). Written informed consent for the survey was obtained from each respondent at the start of each interview. For participants aged between 15 to 17 years, written informed consent from their legally acceptable representative (parent/guardian) was obtained in addition to their assent to take part in the study, unless they were regarded as emancipated minors (i.e., living independently from their legal guardians).

## Results

### Sample characteristics

Of the 10,011 individuals approached between 19 Oct 2021 and 28 April 2023 (8,011 and 2,000 in Karonga and Lilongwe, respectively), 9,849 individuals were recruited representing a 98% response rate (7,930 and 1,919 from Karonga and Lilongwe, respectively). Table 1 lists the demographic and health-related characteristics of the study population by sex and site. Overall, 56% of the participants were female. The overall median age of the total population was 30.0 years (Inter Quartile Range, 21.0–43.0). The mean age was higher among women than men in the total population 34.9±16.2 vs 33.2±16.4 years, $p<0.001$, and in the rural population (35.6 ±16.9 vs33.6±16.8) $p<0.001$. Of the participants, 51% had not completed any formal education, and 50% were self-employed or had irregular employment. In terms of lifestyle factors, a majority (92%) reported to have never smoked, and 19% had not consumed alcohol in the past 12 months. Among the participants, 17% met criterion for being physically active, most (67%) had normal weight while 25% were overweight/obese with women having the highest prevalence, at 33% (Table 1). We excluded 11% (1,095 participants) from the main analysis because they had missing data on multimorbidity.

We observed significant differences in mean depressive and anxiety symptom scores in the considered sociodemographic and health-related factors as displayed in S1 Appendix. Specifically, we observed an increase in the mean depressive symptom scores as the number of physical conditions increased, with a higher mean score among individuals with three or more physical health conditions. Additionally, there was a strong correlation between depression and anxiety scores (Spearman correlation coefficient ($\rho$) = 0.68).

### Prevalence of long-term conditions and multimorbidity

Fig 1 presents the age-standardized prevalence of physical health conditions and multimorbidity of these conditions. The age-standardized prevalence of hypertension was significantly higher among urban (24.3%, 95%CI = 22.6–26.0) compared to rural (15.1%, 95%CI = 14.4–15.8) participants. The age-standardized prevalence of chronic pain was significantly higher in rural participants (9.3%, 95%CI = 8.7–9.9) as compared to urban participants (6.8%, 95% CI = 5.6–8.1). There were no significant differences in the age-standardized prevalence of multimorbidity in urban participants (14.1%,95% CI, 12.5–15.8) and rural individuals (12.2%, 95% CI, 11.6–12.9). Multimorbidity prevalence increased with age, but no significant differences were observed between males from urban and rural areas, and females from urban and rural areas (Fig 2).

### Common physical health conditions combinations

The most common disease combinations among individuals with two conditions were hypertension and disability (18%), and hypertension and chronic pain (14%). Among those with three conditions, the most prevalent combinations were hypertension, disability, and chronic pain (24%), and hypertension, disability, and HIV (8%). Diabetes, hypertension, disability, and chronic pain were the most common combination (23%) among those with four conditions (S2 Appendix).

### Association between physical health multimorbidity and common mental health disorders

Table 2 presents the association between physical health multimorbidity and depression and anxiety scores. After adjusting for covariates, depressive symptom scores significantly

**Table 1. Characteristics of participants in rural and urban sites.**

| Variable | | Total (n = 9849) | | Karonga (rural, n = 7930) | | Lilongwe (urban, n = 1919) | |
|---|---|---|---|---|---|---|---|
| | All (n = 9849) | Men (n = 4287) | Women (n = 5562) | Men (n = 3501) | Women (4429) | Men (n = 786) | Women (n = 1133) |
| *Sociodemographic factors* | | | | | | | |
| Median age, IQR | 30.0 (21.0–43.0) | 28.0 (20.0–43.0) | 31 (22.0–44.0) | 28.0 (20.0–44.0) | 32.0 (22.0–45.0) | 27.0 (21.0–39.0) | 29.0 (22.0–39.0) |
| Mean age, SD (in years)[a] | 34.1 (16.3) | 33.2 (16.4) | 34.9 (16.2) | 33.6 (16.8) | 35.6 (16.9) | 31.3 (14.1) | 32.0 (12.8) |
| Age group (years) | | | | | | | |
| 15–17 | 1142 (12%) | 590 (14%) | 552 (10%) | 505 (14%) | 459 (10%) | 85 (11%) | 93 (8%) |
| 18–29 | 3743 (38%) | 1708 (40%) | 2035 (37%) | 1328 (38%) | 1549 (35%) | 380 (48%) | 486 (43%) |
| 30–39 | 1906 (19%) | 720 (17%) | 1186 (21%) | 581 (17%) | 910 (21%) | 139 (18%) | 276 (24%) |
| 40–49 | 1383 (14%) | 569 (13%) | 814 (15%) | 484 (14%) | 651 (15%) | 85 (11%) | 163 (14%) |
| 50–59 | 782 (8%) | 345 (8%) | 437 (8%) | 288 (8%) | 363 (8%) | 57 (7%) | 74 (7%) |
| ≥60 | 893 (9%) | 355 (8%) | 538 (10%) | 315 (9%) | 497 (11%) | 40 (5%) | 41 (4%) |
| Education attainment | | | | | | | |
| Not completed any level | 5024 (51%) | 1887 (44%) | 3137 (56%) | 1722 (49%) | 2777 (63%) | 165 (21%) | 360 (32%) |
| Primary | 1604 (16%) | 745 (17%) | 859 (15%) | 601 (17%) | 645 (15%) | 144 (18%) | 214 (19%) |
| Junior secondary | 1342 (14%) | 616 (14%) | 726 (13%) | 494 (14%) | 500 (11%) | 122 (16%) | 226 (20%) |
| Senior secondary | 1175 (12%) | 693 (16%) | 482 (9%) | 484 (13%) | 295 (7%) | 209 (27%) | 187 (17%) |
| Post secondary | 539 (5%) | 303 (7%) | 233 (4%) | 172 (5%) | 117 (7%) | 134 (17%) | 116 (10%) |
| Missing | 165 (2%) | 40 (1%) | 125 (2%) | 28 (1%) | 95 (2%) | 12 (2%) | 30 (3%) |
| Marital status | | | | | | | |
| Never married | 2942 (30%) | 1790 (42%) | 1152 (21%) | 1373 (39%) | 851 (19%) | 417 (53%) | 301 (27%) |
| Married | 5431 (55%) | 2246 (52%) | 3185 (57%) | 1914 (55%) | 2527 (57%) | 332 (42%) | 658 (58%) |
| Divorced/separated | 902 (9%) | 207 (5%) | 695 (13%) | 179 (5%) | 584 (13%) | 28 (4%) | 111 (10%) |
| Widowed | 574 (6%) | 44 (1%) | 530 (10%) | 35 (1%) | 467 (11%) | 9 (1%) | 63 (6%) |
| Employment | | | | | | | |
| Not employed | 2696 (27%) | 741 (17%) | 1955 (35%) | 575 (16%) | 1470 (33%) | 166 (21%) | 485 (43%) |
| Full-time student | 1486 (15%) | 817 (19%) | 669 (12%) | 661 (19%) | 506 (11%) | 156 (20%) | 163 (14%) |
| Employed/regular | 704 (7%) | 448 (11%) | 256 (5%) | 263 (8%) | 146 (3%) | 185 (24%) | 110 (10%) |
| Self/irregular | 4963 (50%) | 2281 (53%) | 2682 (48%) | 2002 (57%) | 2307 (52%) | 279 (36%) | 375 (33%) |
| *Health-related factors* | | | | | | | |
| Smoking status | | | | | | | |
| Never | 9097 (92%) | 3561 (81%) | 5536 (99%) | 2907 (83%) | 4416 (99%) | 654 (83%) | 1120 (99%) |
| Former | 314 (3%) | 297 (7%) | 17 (<1%) | 228 (7%) | 7 (<1%) | 69 (9%) | 10 (<1%) |
| Current | 438 (5%) | 429 (10%) | 9 (<1%) | 366 (10%) | 6 (<1%) | 63 (8%) | 2 (<1%) |
| Alcohol consumption | | | | | | | |
| Not in last year | 8018 (81%) | 2680 (63%) | 5338 (96%) | 2164 (62%) | 4265 (96%) | 516 (66%) | 1073 (95%) |
| In last year | 1827 (19%) | 1604 (37%) | 223 (4%) | 1337 (38%) | 164 (4%) | 267 (34%) | 59 (5%) |
| Missing | 4 (<1%) | 3 (<1%) | 1 (<1%) | - | - | 3 (<1%) | 1(<15) |
| Physical activity | | | | | | | |
| Not met strength guidelines | 8052 (82%) | 3005 (70%) | 5047 (91%) | 2486 (71%) | 4026 (91%) | 519 (66%) | 1021 (90%) |
| Met strength guidelines | 1660 (17%) | 1196 (28%) | 464 (8%) | 936 (27%) | 355 (8%) | 260 (33%) | 109 (10%) |
| Missing | 137 (1%) | 86 (2%) | 51 (1%) | 79 (2%) | 48 (1%) | 7 (<1%) | 3 (<1%) |
| Body mass index (kg/m$^2$) | | | | | | | |
| Underweight | 692 (7%) | 426 (10%) | 226 (5%) | 358 (10%) | 213 (5%) | 68 (9%) | 53 (5%) |
| Normal | 6614 (67%) | 3338 (78%) | 3276 (59%) | 2752 (79%) | 2680 (61%) | 586 (75%) | 596 (53%) |
| Overweight | 1646 (17%) | 396 (9%) | 1250 (22%) | 299 (9%) | 995 (22%) | 97 (12%) | 255 (23%) |
| Obese | 820 (8%) | 98 (9%) | 722 (13%) | 68 (2%) | 505 (11%) | 30 (4%) | 217 (19%) |
| Missing | 77 (<1%) | 29 (<1%) | 48 (<1%) | 24 (<1%) | 36 (<1%) | 5 (<1%) | 12 (1%) |

*(Continued)*

**Table 1.** (Continued)

| Variable | | Total (n = 9849) | | Karonga (rural, n = 7930) | | Lilongwe (urban, n = 1919) | |
|---|---|---|---|---|---|---|---|
| | All (n = 9849) | Men (n = 4287) | Women (n = 5562) | Men (n = 3501) | Women (4429) | Men (n = 786) | Women (n = 1133) |
| Number of physical health conditions [b] | | | | | | | |
| None | 5657 (57%) | 2637 (62%) | 3020 (54%) | 2169 (62%) | 2423 (55%) | 468 (60%) | 597 (53%) |
| One | 1865 (19%) | 679 (16%) | 1186 (21%) | 579 (17%) | 979 (22%) | 100 (13%) | 207 (18%) |
| Two | 912 (9%) | 312 (7%) | 600 (11%) | 244 (7%) | 481 (11%) | 68 (9%) | 119 (11%) |
| Three or more | 320 (3%) | 94 (2%) | 226 (4%) | 83 (2%) | 194 (4%) | 11 (1%) | 32 (3%) |
| Incomplete data | 1095 (11%) | 565 (13%) | 530 (10%) | 426 (12%) | 352 (8%) | 139 (18%) | 178 (16%) |

IQR = Interquartile Range, SD = standard deviation, kg = kilograms, m = meters

[a] p-values for difference in mean age between men and women, total population <0.001, rural site <0.001, and urban site = 0·203, results from t-test

Body mass index defined as underweight ($<18.0$ kg/m$^2$), normal weight (18.0–24.9 kg/m$^2$), overweight (25.0–29.9 kg/m$^2$) and obese ($\geq30$ kg/m$^2$)

[b] Physical health conditions included diabetes, hypertension, heart disease, stroke, chronic pain, disability, epilepsy, HIV, and asthma. Unadjusted prevalence of multimorbidity

increased with an increasing number of physical health conditions. Compared to those who had no physical health condition, depressive symptom score (y) increased for each increment increase in physical health conditions (x): one condition (Coefficient (B) 0.78; 95% CI 0.63–

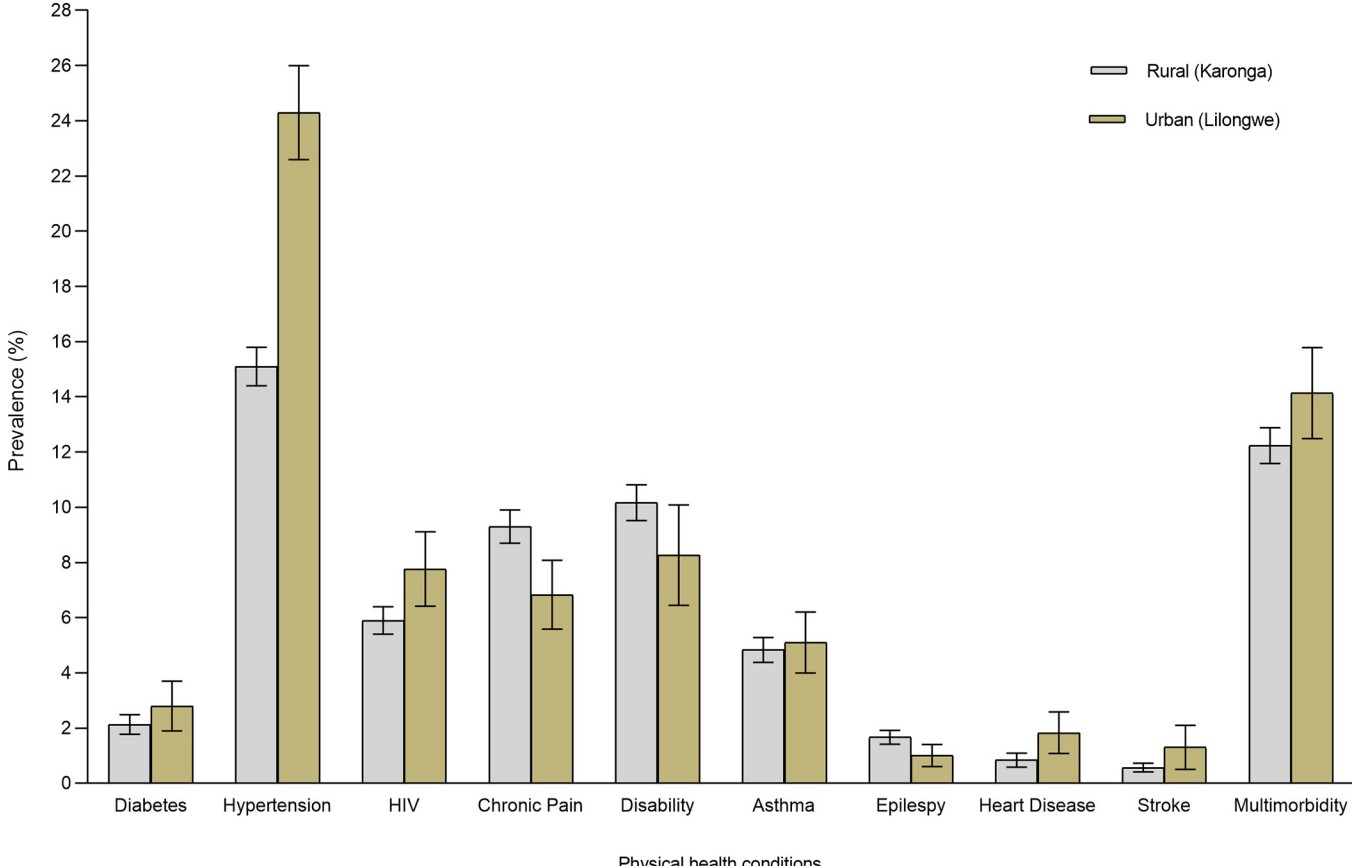

**Fig 1. Age-standardized prevalence of physical health conditions by site.** Diabetes and hypertension included objective measures plus self-reports while the rest of the conditions were self-reported.

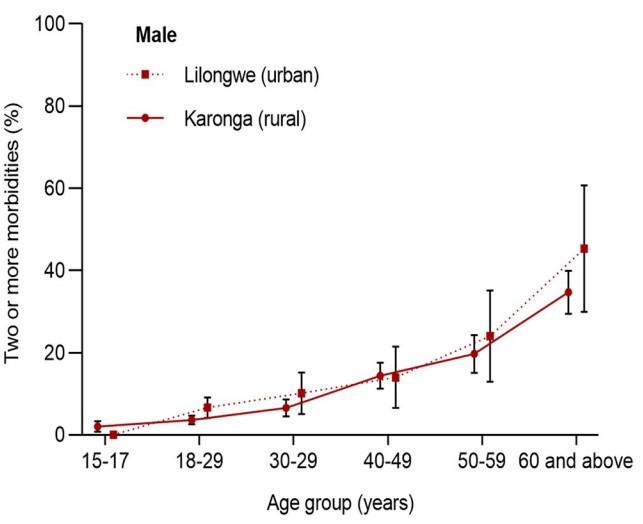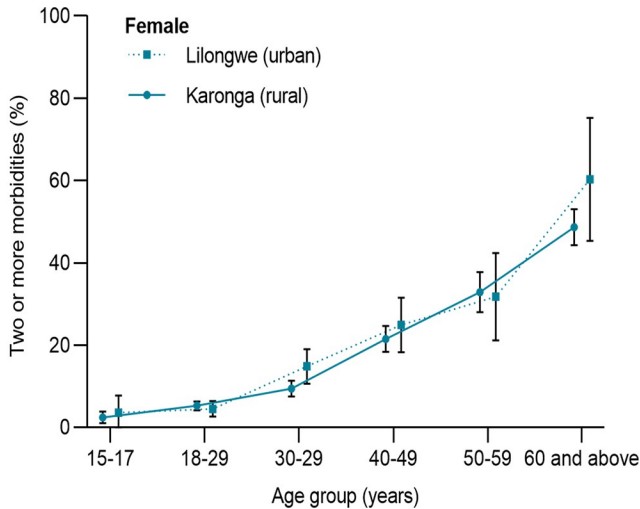

**Fig 2. Age- and sex-specific prevalence of multimorbidity by site.**

0.95), two conditions (*B* 1.36; 95% CI 1.14–1.58), and three or more conditions (*B* 2.23; 95% CI 1.87–2.59), *p*-trend <0.001. Similarly, compared to those without physical health conditions, an increase in anxiety symptom score was observed for each increment increase in physical health conditions; one condition (*B* 0.67; 95% CI 0.53–0.81), two conditions (*B* 1.15; 95% CI 0.96–1.25), and three or more conditions (*B* 1.58; 95% CI 1.27–1.90), *p*-trend <0.001, with adjustment for covariates. While the interaction of sex or site with the association between physical multimorbidity and mental health conditions were not significant, the interaction terms for age and physical multimorbidity were significant but the stratified analysis by age yielded no different results (S3 Appendix).

**Table 2. Association between number of physical health conditions and depression and anxiety scores.**

| | | | | Depression score [a] | | | |
|---|---|---|---|---|---|---|---|
| | | **Unadjusted Model** | | **Model I** | | **Model II** | |
| Number of physical conditions | | B coefficient (95% CI) | P-value | B coefficient (95% CI) | P-value | B coefficient (95% CI) | P-value |
| | None | Ref. | | Ref. | | Ref. | |
| | One | 0.64 (0.48–0.79) | <0.001 | 0.78 (0.63–0.94) | <0.001 | 0.79 (0.63–0.94) | <0.001 |
| | Two | 1.04 (0.84–1.25) | <0.001 | 1.34 (1.12–1.56) | <0.001 | 1.36 (1.14–1.58) | <0.001 |
| | Three or more | 1.57 (1.23–1.91) | <0.001 | 2.16 (1.81–2.51) | <0.001 | 2.23 (1.86–2.59) | <0.001 |
| | *p* value for trend | <0.001 | | <0.001 | | <0.001 | |
| | | | | Anxiety score [b] | | | |
| Number of physical conditions | | | | | | | |
| | None | Ref. | | Ref. | | Ref. | |
| | One | 0.47 (0.32–0.63) | <0.001 | 0.67 (0.53–0.81) | <0.001 | 0.67 (0.53–0.81) | <0.001 |
| | Two | 0.85 (0.63–1.07) | <0.001 | 1.14 (0.94–1.33) | <0.001 | 1.16 (0.97–1.36) | <0.001 |
| | Three or more | 1.01 (0.76–1.45) | <0.001 | 1.52 (1.20–1.83) | <0.001 | 1.59 (1.27–1.91) | <0.001 |
| | *p* value for trend | <0.001 | | <0.001 | | <0.001 | |

[a] Patient Health Questionnaire-9 depression score, measured on continuous scale from 0–27.

[b] General Anxiety Disorder-7 anxiety score, measured on a continuous scale from 0–21.

Model I: Adjusted for sex, age, site education, and employment status, marital status.

Model II: Adjusted for the variables in Model I plus physical activity, BMI, alcohol status, and smoking status.

### Sensitivity analyses

We compared the sociodemographic factors and the outcomes between those that had multi-morbidity data and those with incomplete multimorbidity data, and observed evidence of differences in sociodemographic factors (sex, age, site, education attainment and employment status), but no evidence of differences in the mean depressive and anxiety scores between the two groups (S4 Appendix). Furthermore, effect estimates did not change materially after exclusion of individuals with incomplete data (on any condition included in the multimorbidity definition) (S5 Appendix). Hence, all individuals were included in the final analyses.

## Discussion

These preliminary data, from a large, ongoing cross-sectional study in urban and rural Malawi, show a link between multimorbidity from physical health conditions and increased depression and anxiety symptoms. Notably, the greater the number of co-morbid physical health conditions, the greater the depression and anxiety symptom scores. We observed multimorbidity in 16% of young adults ($< 30$ years) in rural and urban areas, which is comparable to the 13% reported in South Africa within the same age group [56]. The most prevalent long-term conditions in young adults were asthma and chronic pain, followed by disability, hypertension, HIV and epilepsy, with hypertension and disability the most common combination. Our findings suggest that, in the future, a sizeable proportion of the Malawian adult population will be living with multiple long-term health conditions.

In these early data, there is a suggestion of differences in prevalence of multimorbidity between urban and rural dwellers, consistent with the pattern observed in rural and urban India (5.8%; 95% CI 5.7–6.0% for rural and 9.7%; 95% CI 9.4–19.1% for urban, respectively) [57], albeit with materially higher multimorbidity estimates in Malawi. The observed multi-morbidity differences were driven mostly by the higher prevalence of hypertension in urban Malawi, as previously described [42]. With nearly 80% of the Malawian population living in rural locations [58] where health services (including diagnostics and treatments) and clinical expertise is limited, the high rural burden of multimorbidity poses a challenge for the design and delivery of health care needed to sustain the health of the population.

In our study, we observed a strong positive correlation ($\rho = 0.68$) between scores on the PHQ-9 and GAD-7 scales. This finding is similar to previous research conducted in the United Kingdom and Bangladesh, which reported correlations of 0.65 and 0.75, respectively, between these two scales [59–61]. These results suggest PHQ-9 and GAD-7 may have broad application across different settings with inherent sociodemographic differences. The strong correlation between scores also reflects the common co-existence of these conditions [62, 63], overlapping symptoms [64] and shared underlying neurobiological and psychological mechanisms [65, 66].

We noted a "dose response" increase in depression and anxiety symptoms with an increasing number of physical conditions, that did not differ between urban and rural populations, and which is consistent with previous research in other settings, including a study conducted in adults aged 50 years and above in LMIC [67]. Further, findings from a nationally representative survey in India have shown increased odds of having depressive symptoms with an increasing number of chronic conditions [22]. A systematic review on multimorbidity and depression that included studies from LMIC reported a 45% increase in the likelihood of having a depressive disorder with each additional chronic physical condition, compared to individuals without any chronic physical conditions [68]. Similarly, a study encompassing 42 countries (including LMIC) [69] showed that compared to individuals with no physical health condition, the odds of anxiety symptoms were two-fold and five-fold higher in individuals with one condition and five or more conditions, respectively.

In the current study we cannot elucidate the biological mechanisms underpinning the observed associations between increasing number of long-term physical health conditions and higher self-reported depression and anxiety symptoms. In future studies we plan to investigate the role of inflammatory biomarkers such as C-reactive protein which have been shown to predict long-term conditions such as heart disease, hypertension and diabetes [70] and may also increase susceptibility to genetic and environmental stressors, which may subsequently contribute to the onset of depressive symptoms [71, 72]. Polypharmacy, which is common in multimorbid individuals [63], has also been linked with depression [73, 74]. The need for ongoing adherence to treatment programs for multiple conditions [75], the detrimental impacts on daily activities [76] and increased social isolation [77–79] experienced with multimorbidity, are additional recognised risk factors for depression. Similarly, fear and worry associated with managing various diseases and possible lack of treatment or treatment inefficacy has also been linked with increased anxiety [80]. We have previously shown that the burden of "lack of treatment" in underserved populations in Malawi is a key factor in wellbeing in those suffering from long-term conditions [81], and out-of-pocket costs of care will exacerbate that anxiety, hence it is plausible that those with multiple long term conditions who, due the fragmented nature of primary care services and need to attend multiple clinics to access appropriate care, may be disproportionately affected compared to those with single diagnosed conditions.

Hypertension, disability, and chronic pain were the most common combination of multimorbidity in our setting. People living with disability and chronic pain may experience challenges in daily living [82], including their ability to work and socialize [83], which may contribute to poor mental health and which may be exacerbated by limited access to appropriate treatment and support [84]. Chronic pain has been shown to negatively impact sleep quality and to increase stress levels, which may also contribute to the development of depression [85]. Specifically, an earlier study in rural Malawi reported a significant correlation between chronic pain and poor mental health, as measured using PHQ-9 and GAD-7 [86]. To better understand the extent to which modifiable lifestyle determinates differentially impact the risk for long-term health conditions in rural and urban African populations, it is necessary to have detailed and accurate physical activity, sleep, and diet data from large population-based studies. Some of these data will become available from the current study once data collection is complete.

The strength of this research includes the availability of standardised, detailed, and comparable data from urban and rural settings across the adult population aged 15 years and older.

Limitations include the cross-sectional design and potential for reverse causation in the observed associations. To determine causal relationships and the extent to which the relationship between physical multimorbidity and common mental health disorders is bidirectional, will require long term follow-up (which will become available in future), and scheduled ascertainment of time changing data, including lifestyle and physical and mental health conditions. The definition of multimorbidity was limited to the conditions available in our datasets and did not include all long-term adult health conditions in Malawi, such as chronic obstructive pulmonary disease [87].

We observed some differences in the sociodemographic factors of our participants and those of the wider HDSS adult populations which may affect the generalisability of our findings and highlights the importance of using alternative strategies to improve recruitment in these groups for the ongoing study. We excluded individuals with incomplete data regarding their health conditions, except for those cases where we could categorize them based on available data as having either two or three or more physical health conditions. This approach might have resulted in some misclassification and an underestimation of the strength of the observed associations. Nevertheless, observed associations did not change materially following

exclusion of these individuals in sensitivity analyses, suggesting that the effect of misclassification was minimal.

Lastly, it is important to note that these analyses were conducted on a subset of the study. In future analyses, when larger, more population-representative data become available, we will be adequately powered to investigate associations stratified by age-, sex-, and urban-rural location and to investigate the impact of knowledge of multimorbidity on mental health by disaggregating those with a prior diagnosis of conditions (such as diabetes or hypertension) from those who are newly diagnosed during the survey. This will allow us to gain deeper insights into the unique challenges faced by each group and their respective impacts on well-being.

Our early findings highlight the need for a holistic approach to patient care in rural and urban Malawi to address the substantive proportion of adolescents and adults living with multiple physical health conditions and poor mental health. This need, considering Malawi's constrained resources, poses a significant challenge for health policy makers to design and deliver an effective and sustainable health system. Possible approaches include; clinical and nursing training curricula that address multimorbidity scenarios [88], adequately equipping healthcare professionals to support management of a variety of long-term conditions (including common mental disorders) [89] and streamlining healthcare processes toward holistic care [90] through provision of integrated services for the treatment of commonly co-occurring condition, including self-management programmes. In Malawi, mental health services within primary care settings [91], training health-care workers in mental health screening and management, establishing referral pathways to mental health lay counsellors or clinical specialists, and availability of psychosocial support services, are not routine practices but are urgently needed.

Interventions that are scalable and sustainable in low resource environments and acceptable to the population are essential if we are to address the current challenges associated with accessing multiple treatments and services in Malawi. There is a pressing need for interventions, along with effective policies for disease prevention and the promotion of healthy living. These include improving the coordination of care for the elderly population [82], who face the highest burden of multimorbidity, tobacco and sugar taxation, regulated food and beverage advertising, salt reduction strategies, improved public infrastructure to facilitate physical activity, school- and workplace-based programs promoting healthy lifestyles, as well as mental health and well-being, and early identification and management of health conditions [83].

## Conclusion

Our findings highlight the burden of common mental health disorders in individuals living with multiple long term physical health conditions. Efforts are urgently needed to design and deliver effective, holistic health care for people living with multiple physical and mental health conditions in rural and urban Malawi.

## Supporting information

**S1 Appendix. Association between sociodemographic and lifestyle factors and depression and anxiety scores.**
(DOCX)

**S2 Appendix. Common combinations of multiple health conditions.**
(DOCX)

**S3 Appendix. Sensitivity analysis of the association between number of physical health conditions and depression and anxiety scores, stratified by age.**
(DOCX)

**S4 Appendix. Distribution of sociodemographic characteristics by availability of multi-morbidity data.**
(DOCX)

**S5 Appendix. Sensitivity analysis of the association between number of physical health conditions and depression and anxiety scores, excluding those with missing data on any condition included in the multimorbidity definition.**
(DOCX)

## Acknowledgments

The authors express their sincere gratitude to the participants for generously giving their time and cooperation. We would also like to extend our appreciation to the Healthy Lives field interviewers and MEIRU laboratory staff for their unwavering dedication and invaluable contributions to this ongoing work.

For the purpose of open access, the author(s) has applied a Creative Commons Attribution (CC BY) licence to any Author Accepted Manuscript version arising from this submission.

## Author Contributions

**Conceptualization:** Owen Nkoka, Amelia C. Crampin, Alison J. Price.

**Data curation:** Owen Nkoka, Estelle McLean, Amelia C. Crampin, Alison J. Price.

**Formal analysis:** Owen Nkoka, Amelia C. Crampin, Alison J. Price.

**Funding acquisition:** Amelia C. Crampin, Alison J. Price.

**Investigation:** Owen Nkoka, Amelia C. Crampin, Alison J. Price.

**Methodology:** Owen Nkoka, Shekinah Munthali-Mkandawire, Kondwani Mwandira, Providence Nindi, Albert Dube, Innocent Nyanjagha, Angella Mainjeni, Jullita Malava, Abena S. Amoah, Estelle McLean, Robert C. Stewart, Amelia C. Crampin, Alison J. Price.

**Project administration:** Owen Nkoka, Shekinah Munthali-Mkandawire, Kondwani Mwandira, Providence Nindi, Albert Dube, Innocent Nyanjagha, Angella Mainjeni, Jullita Malava, Abena S. Amoah, Robert C. Stewart, Amelia C. Crampin, Alison J. Price.

**Resources:** Amelia C. Crampin, Alison J. Price.

**Supervision:** Owen Nkoka.

**Visualization:** Owen Nkoka.

**Writing – original draft:** Owen Nkoka, Amelia C. Crampin, Alison J. Price.

**Writing – review & editing:** Owen Nkoka, Shekinah Munthali-Mkandawire, Kondwani Mwandira, Providence Nindi, Albert Dube, Innocent Nyanjagha, Angella Mainjeni, Jullita Malava, Abena S. Amoah, Estelle McLean, Robert C. Stewart, Amelia C. Crampin, Alison J. Price.

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
