## [Decision Letter · Decision Letter 0]

3 Jan 2024

PGPH-D-23-02092

Association between physical multimorbidity and common mental health disorders in rural and urban Malawian settings: preliminary findings from Healthy Lives Malawi long-term conditions survey

Dear Dr. Nkoka,

Thank you for submitting your manuscript to PLOS Global Public Health. After careful consideration, we feel that it has merit but does not fully meet PLOS Global Public Health’s publication criteria as it currently stands. Therefore, we invite you to submit a revised version of the manuscript that addresses the points raised during the review process.

We look forward to receiving your revised manuscript.

Kind regards,

Joel Msafiri Francis, MD, MS, PhD

Academic Editor

Journal Requirements:

Additional Editor Comments (if provided):

Reviewers' comments:

Reviewer's Responses to Questions

**Comments to the Author**

1. Does this manuscript meet PLOS Global Public Health’s publication criteria? Is the manuscript technically sound, and do the data support the conclusions? The manuscript must describe methodologically and ethically rigorous research with conclusions that are appropriately drawn based on the data presented.

Reviewer #1: Yes

Reviewer #2: Yes

2. Has the statistical analysis been performed appropriately and rigorously?

Reviewer #1: Yes

Reviewer #2: Yes

3. Have the authors made all data underlying the findings in their manuscript fully available (please refer to the Data Availability Statement at the start of the manuscript PDF file)?

Reviewer #1: Yes

Reviewer #2: Yes

4. Is the manuscript presented in an intelligible fashion and written in standard English?

Reviewer #1: Yes

Reviewer #2: Yes

5. Review Comments to the Author

Reviewer #1: Reviewer Comments:

Abstract

1. Lines 22-24:

Do the authors mean to add the word “conditions” at the end?

“Multimorbidity was defined as having two or more measured (diabetes, hypertension) or self-reported (diabetes, hypertension, disability, chronic pain, HIV, asthma, stroke, heart disease, and epilepsy).”

2. Line 32:

“hypertension, and disability (18%) co-occurred most frequently,” the “18%” is better placed at the end of the word “frequently?”

3. Line 33:

“hypertension, disability, and chronic pain (23%),” the “23%” is better placed at the end of the word “combination?”

4. Lines 39-40:

Reads better: “and need to integrate mental and physical health service delivery in Malawi.

Introduction

1. Line 53:

Kindly add the reference after the multimorbidity definition. “Multimorbidity (MM), defined as the presence of two or more long-term conditions [ref],”

2. Lines 55-56:

Is the mentioned 60% multimorbidity prevalence with reference to young adults or the general population within LMICs?

Please specify the prevalence in young adults (<25 years) and also older adults ((≥65 years).

3. Lines 63-64:

Kindly add references after this statement: “Previous studies in LMIC have reported links between mental health disorders and multimorbidity.”

Methods

1. Line 93:

Perhaps start the sentence with “The” instead of “Our”

2. Line 110:

Please define what “usual members” are.

3. Line 116:

Kindly explain the significance of the thumbprint consent. Was this for participants who were unable to give written consent because they were unable to read or write?

4. Line 123:

Should be “(PHQ-9)” and not “(PHQ)-9”

5. Lines 127-128:

Are there categories associated with the said scores? If so, please describe them instead of saying “with higher scores corresponding to more symptoms and increasing severity of depression.”

6. Lines 132-133:

Are there categories associated with the said values? If so, please describe them instead of saying “with higher values indicating more symptoms and more severe anxiety.”

7. Line 137:

Kindly add reference after the multimorbidity definition.

8. Overall:

Despite ongoing enrolment and data collection at the study sites, the methods (what was done for purposes of this particular study) need to be written in the past tense. For example, Lines 110-111 “Individuals who are aged 15 years and above and who are usual members/residents of households (in the rural or urban HDSS) are eligible for recruitment.”

Results

1. Overall:

Results are explained in detail, appropriate statistical tests used, and thanks to the authors for conducting the sensitivity analyses!

Discussion

1. Line 288:

The words: “for urban” missing after “95% CI 9.4 – 19.1%.”

2. Lines 295-296:

Kindly provide more information on these mentioned “previous findings” – where was the study conducted? Are the sociodemographic factors such as age, sex, etc similar to your study?

3. Lines 303-304:

What was the systematic review on? Was it conducted on studies in LMICs as well?

4. Line 305:

Were these 42 countries LMICs too?

5. Overall:

Perhaps provide comparisons with studies conducted in other LMICs other than India and South Africa only.

Overall:

Well done to the authors for writing such a relevant manuscript with appropriate statistical methods. It has been an informative and interesting read.

Reviewer #2: Manuscript number: PGPH-D-23-02092

Journal: Plos Global Public Health

Thank you for the opportunity to review the manuscript titled “Association between physical multimorbidity and common mental health disorders in rural and urban Malawian settings: preliminary findings from Healthy Lives Malawi long-term conditions survey” on behalf of PloS Global Public Health.

Multimorbidity is an area of increasing concern. As the authors state, there is very little information available about multimorbidity in low and middle income (LMIC) contexts, and even less so in low income contexts.

I would like to commend the authors on a thorough, clearly described and well written paper. The authors make a good argument for why country-specific studies on multimorbidity are needed and the paper’s focus on multimorbidity in Malawi is welcome. The additional focus on mental health is also welcome as this is known to be an underserviced area in many LMICs.

The Methods are well described and the authors make use of a rich dataset, which can be difficult to find in LMIC contexts; thereby relaying important results about this population.

The results are well laid out. The finding of 16% prevalence in young people in Malawi is quite interesting and perhaps unexpected.

The paper is important and provides baseline information from a large dataset in Malawai. I feel that it is also highly relevant to the journal and the collection of articles (multimorbidity in Africa).

I would advise in future that the authors consider conducting a latent class analysis or cluster analysis to interpret the disease groupings. I look forward to seeing more research come out of this study.

Minor

Abstract

- Line 23 is missing the word “disease” or “condition”.

Methods

- Line 110: Although I know why there is a focus on individuals 15 years and older, in many of my own articles I have been asked by reviewers to explain why this age group starts at 15 years rather than 18 years. I think it would be good to explain the significance of starting the observations with people aged 15 years.

- Line 120: Was there a reason that data collection between the two sites did not occur at the same time instead of a year apart?

- Line 134: Have the tools been validated in Malawi before? If not, I think your validation of the tool deserves to be highlighted more.

- Line 147 – 150: Please reference the standards that were used to determine the cut-offs for diabetes and hypertension.

- Line 195: Please fix the sentence, it should be for people aged 15 to 17 years of age. “For participants aged between under 18 years, written informed consent from their legally acceptable representative”

Results

- I appreciate that Table 1 is quite a busy table. However, could the places where p-values were significant be highlighted in some way. I note that it is stated in a footnote however I would like to be able to look at the table and be able to see it immediately.

Discussion

- Are the authors able to identify what diseases are driving the high prevalence of multimorbidity in young people in Malawi?

6. PLOS authors have the option to publish the peer review history of their article (what does this mean?). If published, this will include your full peer review and any attached files.

**Do you want your identity to be public for this peer review?** For information about this choice, including consent withdrawal, please see our Privacy Policy.

Reviewer #1: **Yes: **Mafuno Grace Mpinganjira

Reviewer #2: **Yes: **Dr Rifqah Abeeda Roomaney

---

## [Editor Report · Decision Letter 1]

5 Feb 2024

Association between physical multimorbidity and common mental health disorders in rural and urban Malawian settings: preliminary findings from Healthy Lives Malawi long-term conditions survey

PGPH-D-23-02092R1

Dear Dr Nkoka,

We are pleased to inform you that your manuscript 'Association between physical multimorbidity and common mental health disorders in rural and urban Malawian settings: preliminary findings from Healthy Lives Malawi long-term conditions survey' has been provisionally accepted for publication in PLOS Global Public Health.

Best regards,

Joel Msafiri Francis, MD, MS, PhD

Academic Editor